# Analyzing the Therapeutic Efficacy of Bis-Choline-Tetrathiomolybdate in the *Atp7b*^−/−^ Copper Overload Mouse Model

**DOI:** 10.3390/biomedicines9121861

**Published:** 2021-12-08

**Authors:** Philipp Kim, Chengcheng Christine Zhang, Sven Thoröe-Boveleth, Eva Miriam Buhl, Sabine Weiskirchen, Wolfgang Stremmel, Uta Merle, Ralf Weiskirchen

**Affiliations:** 1Institute of Molecular Pathobiochemistry, Experimental Gene Therapy and Clinical Chemistry (IFMPEGKC), RWTH University Hospital Aachen, D-52074 Aachen, Germany; hkim@ukaachen.de (P.K.); sweiskirchen@ukaachen.de (S.W.); 2Department of Internal Medicine IV, Heidelberg University Hospital, D-69120 Heidelberg, Germany; ChengchengChristine.Zhang@med.uni-heidelberg.de; 3Institute for Occupational, Social and Environmental Medicine, RWTH Aachen University, D-52074 Aachen, Germany; sven.thoroe@rwth-aachen.de; 4Electron Microscopy Facility, Institute of Pathology, RWTH Aachen University Hospital, D-52074 Aachen, Germany; ebuhl@ukaachen.de; 5Medical Center Baden-Baden, D-76530 Baden-Baden, Germany; wolfgangstremmel@aol.com

**Keywords:** Wilson’s disease, *Atp7b*, bis-choline-tetrathiomolybdate, WTX101, ALXN-1840, trientine, D-penicillamine, laser ablation inductively coupled plasma mass spectrometry, therapy, copper

## Abstract

Bis-choline-tetrathiomolybdate, introduced as WTX101 (now known as ALXN1840), is a first-in-class copper-protein-binding agent for oral therapy of Wilson’s disease. In contrast to other decoppering agents such as trientine or D-penicillamine it acts by forming a tripartite complex with copper and albumin, thereby detoxifying excess liver and blood copper through biliary excretion. Preclinical animal experimentation with this drug was typically done with the alternative ammonium salt of tetrathiomolybdate, which is expected to have identical properties in terms of copper binding. Here, we comparatively analyzed the therapeutic efficacy of ALXN1840, D-penicillamine and trientine in lowering hepatic copper content in *Atp7b*^−/−^ mouse. Liver specimens were subjected to laser ablation inductively conductively plasma mass spectrometry and electron microscopic analysis. We found that ALXN1840 caused a massive increase of hepatic copper and molybdenum during early stages of therapy. Prolonged treatment with ALXN1840 reduced hepatic copper to an extent that was similar to that observed after administration of D-penicillamine and trientine. Electron microscopic analysis showed a significant increase of lysosomal electron-dense particles in the liver confirming the proposed excretory pathway of ALXN1840. Ultrastructural analysis of mice treated with dosages comparable to the bis-choline-tetrathiomolybdate dosage used in an ongoing phase III trial in Wilson’s disease patients, as well as D-penicillamine and trientine, did not show relevant mitochondrial damage. In contrast, a high dose of ALXN1840 applied for four weeks triggered dramatic structural changes in mitochondria, which were notably characterized by the formation of holes with variable sizes. Although these experimental results may not be applicable to patients with Wilson’s disease, the data suggests that ALXN1840 should be administered at low concentrations to prevent mitochondrial dysfunction and overload of hepatic excretory pathways.

## 1. Introduction

WTX101, now known as ALXN1840, forms a tight complex with copper (Cu) and Cu-binding proteins with much higher affinities for Cu than other chelators used in therapy of Wilson’s disease such as D-penicillamine (DPA) or trientine (TETA). ALXN1840 reduces plasma nonceruloplasmin-bound Cu by forming tripartite complexes with albumin, targets hepatic intracellular Cu and increases biliary Cu excretion [1,2], while other chelators promote the excretion of Cu via urine. TTM forms a stable, tripartite complex with protein, Cu and itself [3]. The therapeutic efficacy of TTM depends on whether it is given with or without food. Given with food, it interferes with intestinal uptake of Cu. When administered between meals, it binds plasma Cu, and can further remove Cu from metallothionein (MT) and form insoluble Cu complexes deposited in the liver [4]. Due to its high affinity for Cu, it is generally assumed that bis-choline-tetrathiomolybdate (bc-TTM) is therapeutically more effective than other chelators like D-penicillamine or trientine in patients with neuropsychiatric symptoms, as the brain is less able to expel copper than the liver. On the other hand, high doses of TTM are suspected to provoke anemia and liver damage [3]. Although never systematically tested, it has been suggested that these side effects are induced by increased hepatic Cu during treatment and bone marrow depletion of Cu [3].

The efficacy of TTM in preventing acute hepatic Cu-induced injury has been intensively documented in Long-Evans Cinnamon (LEC) rats [5,6]. In line with its mode of action, TTM caused a shift in the chromatographic distribution of Cu, from low molecular weight proteins such as MT to a higher molecular weight protein, most likely albumin [7]. It has been demonstrated that TTM complexes have a strong affinity for Cu [8]. In addition, low concentrations of TTM result in modified MT with unoccupied sulfhydryl groups, which coordinate with Cu intermolecularly to form dimeric and polymeric MT through the –S-Cu-S-bridge, which are then removed from the liver by excretion through feces [7,9,10].

However, excessive TTM doses facilitate polymerization of the Cu/TTM to insoluble polymers that precipitate in the liver, which can only be slowly excreted into the bile and blood [11,12,13]. Liver lysosomes from TTM-treated LEC rats we found to contain lysosomal Cu-Mo-S clusters in which molybdenum (Mo) was coordinated by four sulfurs with approximately three Cu neighbors, suggesting that the formation of these clusters is one critical therapeutic attribute of TTM which is necessary to decrease the bioavailability or redox properties of free Cu in LEC rats [14]. Other LEC studies showed that TTM improves acute hepatitis when applied directly after onset of inflammation, while the drug caused severe hepatotoxicity in one out of 12 rats, albeit in 1.5 h after dosing with fatal outcome [15]. Moreover, depletion of hepatic glutathione (GSH) increased the amount of Cu and Mo deposited in the liver, suggesting that GSH can enhance the therapeutic effect of TTM [16]. Therefore, it is likely that the therapeutic effect and hepatic side-effects of TTM are dose-dependent, as different dosages likely result in different TTM-complexes and routes of excretion.

Unfortunately, the ammonium formulation (a-TTM) is rather unstable, preventing its routine clinical use [17,18]. Therefore, the choline salt ALXN1840 was introduced as a second-generation analogue. In humans, the pharmacokinetic profile of this drug was first tested in a phase I study including 18 patients with advanced solid tumors which were refractory to conventional therapy [17]. In an open-label phase 2 study in which 28 Wilson’s disease patients received ALXN1840 for 24 weeks once daily (NCT02273596), the drug was shown to have a favorable safety profile, and reduced plasma nonceruloplasmin-bound Cu and disease-related disability [1]. In a subsequent ongoing phase 3, randomized, blind, multicenter study, the efficacy and safety of the drug was further tested and compared to standard treatments when administered for 48 weeks (NCT03403205).

Until now, few studies have systematically tested the therapeutic decoppering efficacies of bc-TTM. In rats, the metabolic disposition of bc-TTM, determined as Mo found in plasma, urine, faeces, tissues, test formulation, cage debris, cage wash and carcasses, was markedly different in LEC and Long Evans Agouti (LEA) rats, reflecting the different Cu levels and distribution in LEA and LEC rats [19].

In this study, we tested the efficacy of bc-TTM in reducing hepatic Cu content in *Atp7b*^−/−^ mice, representing a well-established experimental Wilson’s disease model [20,21,22]. Hepatic metal concentrations were determined by laser ablation inductively coupled plasma mass spectrometry (LA-ICP-MS) in liver tissue sections of male homozygous *Atp7b^−/−^* mice that received bc-TTM for 4 or 8 weeks. We found that the drug provoked a significant increase in hepatic Cu and Mo during early phases of therapy (4 weeks), while prolonged treatment (8 weeks) resulted in reduced Cu levels that were similar to those observed after DPA and TETA treatment. Transmission electron microscopy showed that all tested dosages led to an initial accumulation of Cu in the liver, associated with the formation of lysosomal electron-dense particles. Only a high dose of bc-TTM was associated with mitochondrial changes reflected in the formation of leakages with variable sizes.

## 2. Materials and Methods

### 2.1. Compounds

Trientine dihydrochloride (trientine, TETA) and D-pencillamine (DPA) were purchased from Sigma-Aldrich (Taufkirchen, Germany). Bis-choline-tetrathiomolybdate (bc-TTM) was a kind gift of Wilson Therapeutics (Stockholm, Sweden), a company that was acquired by Alexion Pharmaceutical, Inc., Boston, MA, USA) and is now part of AstraZeneca (Gaithersburg, MD, USA). The compounds were dissolved in deionized water at the indicated concentrations.

### 2.2. Animals

The generation of *Atp7b^−/−^* mice used in this study was described in [20]. In brief, gene disruption in these animals was mediated by the insertion of multiple stop codons covering all possible reading frames into exon 2 of the *Atp7b* gene, resulting in a transgenic mice model expressing modified *Atp7b* mRNA, translating into considerably smaller nonfunctional proteins. Mutant mice and age-matched wild type controls were bred on 129/Sv background and housed at the University of Heidelberg under conditions described previously and according to the guidelines of the Institutional Animal Care and Use Committees and all governmental requirements [23]. In this study, a total of 50 male mice (11 wild type, 39 *Atp7b*^−/−^) were analyzed.

### 2.3. Animal Treatment

Two separate experiments were performed. (i) In one experiment (dose finding), *Atp7b*^−/−^ mice (*n* = 9) at age 9 weeks received oral gavages of 1 mg bc-TTM/kg body weight (*n* = 3), 5 mg bc-TTM/kg body weight (*n* = 3), or 10 mg bc-TTM/kg body weight (*n* = 3) dissolved in 100 µL deionized water four times a week for four consecutive weeks. Wild type controls that received 100 µL water per oral gavage at each time point (*n* = 3) or mice sacrificed at the beginning of the experiment (*n* = 3) served as a control. (ii) In the second experiment (therapeutic efficacy), we used a total of 35 animals at age 36 weeks. The wild type animals (*n* = 5) were sacrificed at the beginning of the experiments. The *Atp7b*^−/−^ mice (*n* = 30) were grouped into six groups with each five animals. One group was sacrificed at the beginning of the experiment, while the others were sacrificed at week 44 after receiving either 100 µL water, 1 mg bc-TTM/kg bw, 5 mg/kg body weight, 200 mg TETA/kg body weight, or 200 mg DPA/kg body weight four times a week per oral gavage for eight consecutive weeks. A summary of all animal groups is given in Appendix A. In compliance with the 3R (Replacement, Refinement, and Reduction) principle implemented authoritatively in the European Union-Directive 2010/63 on the protection of animals used for scientific purposes [24,25], data on hepatic Cu content of untreated *Atp7b*^−/−^ mice were taken from previous measurements published by our laboratories [22,23,26,27,28].

### 2.4. Sample Preparation for LA-ICP-MS Measurements

Liver samples were cryo-cut into 30 μm thick slices with a CM3050S cryomicrotome (Leica Biosystems, Wetzlar, Germany) on −18 °C cryo-chamber temperature and −16 °C object area temperature, and thaw-mounted onto adhesive, grid indexed StarFrost^®^ microscope slides (Knittel Glass, Braunschweig, Germany). Samples were air dried and stored at room temperature until analysis.

### 2.5. LA-ICP-MS Set Up and Measurements

Prior to taking measurements, the mounted tissues were first scanned in a NanoZoomer-SQ digital slide scanner (Hamamatsu Photonics Germany GmbH, Herrsching am Ammersee, Germany). The LA-ICP-MS measurements were performed in a system combining a high performance quadrupole Agilent 8900 ICP-MS (Agilent Technologies, Waldbronn, Germany) and a New Wave NWR213 laser ablating device, (Elemental Scientific, Omaha, NE, USA). Alternatively, an XSeries 2 ICP-MS device from Thermo Scientific (Bremen, Germany) was used in our measurements. Standards for the determination of element concentrations were produced from homogenized tissue spiked with varying concentrations of a standard salt solution. As a surrogate of slice thickness that is relevant for final calculation of concentrations within the tissue, the individual metal intensities were normalized to the average ^13^C ion intensity of the respective samples in each measurement [26]. The parameters used in the measurements were: Rf power input: 1300 W, argon plasma gas flow rate: 15 L/min, nebulizer gas flow rate: 0.95 L/min, helium carrier gas flow rate: 1 L/min, Dwell time: 13 ms, mass resolution: max. 500 m/Δm, scanning mode: peak hopping, laser spot size 60 µm, scan speed: 70 µm/sec, ablation mode: line scan, repetition frequency: 20 Hz, laser fluence: 1.6 J/cm^2^. This results in a typical analysis time of 4–6 h per liver sample. Element concentrations in Figures and Tables showing LA-ICP-MS data indicate the spatial distribution of metals within the tissue.

### 2.6. ICP-MS Measurements

To confirm our LA-ICP-MS measurements, additional ICP-MS measurements of ^63^C, ^65^Cu, Mo, Zn and Fe in 72 mouse serum and 57 mouse liver samples were commissioned by a commercial supplier (Quotient Sciences, Bioanalytical Services, Alnwick, UK). As internal controls, both ^63^Cu and ^65^Cu were measured. These are the only stable isotopes of Cu, occurring with abundances of 69.17% (^63^Cu) and 30.83% (^65^Cu). In proper measurements, the determined concentrations of both isotopes should vary by a factor of ~2.24.

### 2.7. Image Generation of Bio-Metal Distribution

Isotope images were generated in Microsoft Excel using the in-house generated Excel Laser Ablation Imaging (ELAI) visualization tool described previously [29,30]. This software allows the generation of images from mass spectrometry data and is available free of charge [29].

### 2.8. Electron Microscopic Analysis

Electron microscopic analysis of *Atp7b*^−/−^ mouse liver tissue was done essentially as described before [31]. In brief, after dissection of the livers, small tissue samples with a size 1–2 mm were prepared and fixed in 3% glutaraldehyde prepared in 1× phosphate buffered saline (PBS). Samples were washed in 0.1 M Soerensen’s phosphate buffer (Merck, Darmstadt, Germany), post-fixed in 1% Osmium tetroxide (OsO_4_) (Roth, Karlsruhe, Germany) in 17% sucrose buffer (Merck), dehydrated through ascending ethanol series, and embedded in epoxy resin as reported before [31]. Ultrathin sections (70–100 nm) were prepared and spread on Cu/Rh grids (HR23 Maxtaform, Plano GmbH, Wetzlar, Germany). The samples were examined with a Zeiss Leo 906 (Carl Zeiss AG, Oberkochen, Germany) transmission electron microscope operated at 60 kV.

### 2.9. Statistics

The statistics shown in Appendix A were generated using the online One Way ANOVA Calculator (Analysis of Variance, Tukey HSD test with the following settings: α: 0.05, outliers: included; effect size: medium; effect type: f, and effect size value: 0.25 [32].

## 3. Results

### 3.1. Hepatic Copper Accumulation in the Atp7b^−/−^ Mice Model as Demonstrated by Laser Ablation Inductively Coupled Plasma Mass Spectrometry

The *Atp7b^−/−^* mouse carrying a homozygous mutation in the *Atp7b* gene was introduced as a model with which to study hepatic manifestations of Wilson’s disease [20]. In this model, there is a dramatic gradual accumulation of hepatic Cu at a young age, followed by the formation of fibrosis and protruding regenerative cirrhotic nodes in middle- to old-aged mice [20]. Compared to normal control mice, a 60-fold greater increase in hepatic Cu during life span is observed in respective mice, which resembles the human disease condition [20,23]. The model has been used previously to follow aspects of Cu accumulation in liver and brain, to characterize metabolic dysfunction associated with increased Cu, and to evaluate the success of gene therapy correction or pharmaceutical interventions [21,22,23,26,27,33,34,35].

In recent years, we introduced novel LA-ICP-MS imaging protocols to identify and quantify changes in hepatic major, minor and trace metals in thin cryosections of *Atp7b^−/−^* mice [21,22,26,28,31,33]. Absolute concentration of various elements can be done by normalization to surrogate markers such as ^13^C, indicating slice thickness and comparison to a standard, in which the concentration of individual elements are well-defined [36]. In a final step, the measured element concentrations can be depicted as colorful “at a glance” imaging maps using different software tools [30]. As reported before, this method is suitable to demonstrate hepatic overload in *Atp7b*^−/−^ mice (Appendix A). In line with our previous experiments, elevated Cu was associated with additional changes in other elements including sodium, phosphorus, iron, manganese and zinc [22,26]. Most strikingly, Cu content was dramatically lowered in the regenerative nodules, supporting previous data showing loss of hepatic Cu to be associated with ongoing liver tissue damage [20].

### 3.2. Occurrence of Granular Electron-Dense Particles in Livers of Atp7b^−/−^ Mice Model as Demonstrated by Transmission Electron Microscopy

Previous studies have demonstrated hepatic Cu overload to be associated with many pathological changes at an ultrastructural level, including severe mitochondrial changes, increased numbers of peroxisomes, lipid droplet formation, occurrence of lipopolyosomes, and electron-dense lysosomal deposits [37,38,39]. We were able to demonstrate such electron-dense lysosomal deposits and fat droplets in liver specimens of *Atp7b*^−/−^ mice using protocols recently established in our laboratory [31]. In most cases, these granular particles were arranged in lysosomal vesicles irregularly distributed within the tissue (Appendix A).

### 3.3. Therapeutic Efficacy of Different Dosages of Bis-Choline-Tetrathiomolybdate

Cu accumulates throughout the liver tissue of *Atp7b^−/−^* mice during the first months of life span, while at higher age, there is intrahepatic Cu loss due to ongoing tissue damage [20,22,23,26]. To test the therapeutic efficacy of bc-TTM in *Atp7b*^−/−^ null mice, we first performed a dose finding study, in which animals at age 9 weeks were treated with oral gavages of different concentrations of the drug (1, 5, and 10 mg bc-TTM/kg body weight) four times a week for four consecutive weeks. Thereafter, animals were sacrificed and liver specimen subjected to LA-ICP-MS measurement. Unexpectedly, the treatment with bc-TTM in the 1 mg group caused doubling of hepatic Cu concentration from 112.72 ± 13.26 µg/g liver tissue to 203.03 ± 21.91 µg/g liver tissue), while a moderate therapeutic effect in the 5 mg group (106.9 ± 45.65 µg/g) and an 30% reduction of this trace element in the 10 mg bc-TTM group (86.9 ± 15.69 µg/g) was observed (Figure 1).

In addition, we also measured hepatic Cu concentration in ashed samples and serum Cu levels by inductively coupled plasma mass spectroscopy (ICP-MS). Here, we also detected high levels of both ^63^Cu and ^65^Cu in the liver and serum samples of *Atp7b^−/−^* mice (Appendix A). In line with our LA-ICP-MS measurements, in our ICP-MS analysis, we found that hepatic Cu content was significantly elevated in the treatment groups, with the highest concentrations in the 1 mg bc-TTM group (Appendix A). In addition, the quantities of Cu were higher than in the control groups in our dose finding study, demonstrating that the drug applied in short term fixes Cu within the liver. In line with this hypothesis, we found that the concentration of Mo that is indicative for bc-TTM increased in the treatment groups.

The measurement of other elements by LA-ICP-MS and ICP-MS in these specimens revealed that the increased Cu content was further associated with elevated concentrations of zinc in the bc-TTM-treated groups (Appendix A).

A previous clinical study showed that bc-TTM treatment, although generally well tolerated, increased concentrations of asymptomatic alanine aminotransferase (ALT), aspartate aminotransferase (AST) or γ-glutamyltransferase (γ-GT) in 39% of all patients; there are clinical markers of hepatotoxicity and liver damage [1]. Therefore, we next evaluated liver specimens of animals treated with the three different dosages of bc-TTM for potential ultrastructural changes. Transmission electron microscopy showed that livers of 10 mg TTM group showed a significant fraction of mitochondria with structural changes. Most striking was the acquisition of a spherical shape with unique cavities or holes, a phenomenon that was not seen in livers of the 1 mg and 5 mg group (Figure 2).

In addition, we also measured serum ALT levels of these animals and observed significantly higher values in the 1 mg TTM group (*p* < 0.001), while the ALT values in the 5 mg and 10 mg TTM groups were not elevated (cf. Appendix A).

Based on these results, and taking into account potential clinical adverse effects of TTM occurring at high concentrations [40,41,42,43], we decided to use the 5 mg bc-TTM dosing regimen in our next set of experiments, in which we extended the treatment interval to eight weeks and used older animals (36 weeks), which at this age have already developed fully hepatic Cu overload [20,22,23,26]. Additionally, we also included a 1 mg bc-TTM/kg body weight group in our next experiments to clarify if the initial increase in Cu observed in our dose finding study reversed during prolonged treatment periods. To further allow a comparison with clinically established Cu chelators, we incorporated a DPA group and a TETA group, both orally given at concentrations of 200 mg/kg body weight four times a week, which is an established dosage applied in experimental models to achieve a therapeutic effect that is comparable with those obtained in humans [44,45]. After 8 weeks of treatment, the animals were sacrificed and element concentrations in specimens were determined by LA-ICP-MS imaging (Figure 3 and Appendix A).

During therapy with 1 mg bc-TTM, the Cu concentration decreased from 26.93 ± 30.41 µg/g to 10.91 ± 8.03 µg/g liver tissue, while TETA or DPA lowered hepatic Cu concentration of 7.0 ± 2.88 µg/g or 6.47 ± 5.98 µg/g liver tissue (Appendix A). In the 5 mg bc-TTM group, the average concentration of hepatic Cu was higher (16.83 ± 13.86 µg/g liver tissue) because one animal in this group developed tumors enriched in Cu as already reported before [22,23,26]. When omitting this animal, the average value for hepatic Cu was 9.99 ± 8.21 µg/g tissue, a concentration lower than those determined in the 1 mg bc-TTM group, but still slightly higher than in the TETA or DPA groups. Importantly, the treatment with all three chelators provoked a more homogenous distribution within the tissue compared to control *Atp7b*^−/−^ animals that received only gavages of water. However, all chelators failed to reduce hepatic Cu content to a physiological range, which is rather constant during life span ranging about 3.41 ± 0.28 µg/g (at age 9 weeks) or 3.45 ± 0.30 µg/g liver tissue (at age 20 weeks) in wild type controls. In addition, we also measured hepatic Cu concentrations in ashed samples and serum levels by ICP-MS (Appendix A).

Similar to the results described above, a decrease of both ^63^Cu and ^65^Cu was detectable in the 1 mg bc-TTM group from 27,687.2 ± 19,022.81 ng/g to 14,640 ± 10,200.49 ng/g liver wet tissue and 12,081.4 ± 8373.05 ng/g to 6286.4 ± 4368.62 ng/g liver tissue. However, using One Way ANOVA testing, we could demonstrate that the observed differences were not significant, when ANOVA testing was used to compare the groups that received water, 1 mg bc-TTM, or 5 mg bc-TTM (*p* = 0.3519). In the treatment groups, Cu in serum was statistically highly significant elevated when comparing to the group that received only water (Appendix A). Accordingly, the concentration of Mo that is indicative for bc-TTM increased in liver and serum in respective groups.

TETA or DPA lowered both hepatic ^63^Cu and ^65^Cu concentrations to 9109 ± 7816.85 ng/g and 3893.8 ± 3397.01 ng/g liver tissue, or 6049 ± 2954.2 ng/g and 2596.67 ± 1307.88 ng/g liver tissue, respectively. As described above, in the 5 mg bc-TTM group, the average concentration of hepatic ^63^Cu and ^65^Cu was higher (28,163.4 ± 12,389.64 ng/g liver tissue and 12,358.4 ± 5782.14 ng/g liver tissue) because one animal in this group developed tumors enriched in Cu, as reported previously; obviously, values were lower when omitting this animal. In line with our previous findings, hepatic Cu concentrations of *Atp7b*^−/−^ mice at age 36 weeks were higher than those observed in control mice at age of 44 weeks, most likely indicating the ongoing tissue damage (cirrhosis) associated with intrahepatic Cu loss reported before [26].

Serum ^63^Cu concentrations were significantly higher in the 5 mg bc-TTM group (941.8 ± 123 ng/mL) compared to control *Atp7b*^−/−^ animals (528.8 ± 54.83 ng/mL; *p* < 0.001) as well as serum ^65^Cu concentrations (402.2 ± 53.97 ng/mL vs. 227.8 ± 23.39 ng/mL; *p* < 0.001) (*t*-test, *p* = 0.0006), while serum values for Zn and Fe in serum did not differ significantly among each group (Appendix A).

Previous studies have suggested that during TTM treatment, insoluble Cu/TTM are formed precipitating in the liver, which can only be slowly excreted into the bile and blood [11,12,13]. In line with this hypothesis, we found that treatment with bc-TTM is associated with accumulation of Mo within the liver (Figure 4) and significantly elevated hepatic Mo concentrations for the 5 mg bc-TTM group (*p* < 0.001) (Appendix A).

Serum Mo levels were also significantly higher in both 1 mg and 5 mg bc-TTM groups (Appendix A).

Interestingly, livers of animals treated with bc-TTM showed more electro-dense lysosomal deposits in some of the liver cells investigated than animals treated with DPA or TETA (Figure 5).

Next, we evaluated liver specimens of animals treated for potential ultrastructural mitochondrial changes. Transmission electron microscopy revealed that livers of all treatment groups (bc-TTM 1 mg and 5 mg group, trientine, and DPA as well as controls) did not show the ultrastructural changes of mitochondria that had been found in the 10 mg bc-TTM group of the previous experiment (Figure 6).

In addition, we also measured serum ALT levels to detect possible hepatic damage in these animals and could not find any significant differences among the treatment groups compared to control *Atp7b*^−/−^ animals (Appendix A).

## 4. Discussion

Presently, several drugs taken orally are routinely used to treat Wilson’s disease. Zinc, given as zinc acetate, sulfate, or gluconate, inhibits intestinal absorption of Cu by inducing MT formation that blocks Cu absorption from the intestinal track [43]. Most frequently used are Cu-chelating drugs including DPA and TETA. DPA (Cuprimine) is derived from the hydrolytic degradation of penicillin with capacity to chelate accumulated Cu and eliminate it by urinary excretion [46,47]. Adverse reactions to DPA are acute sensitivity reactions, skin rashes, blood dyscrasias, and in some rare cases renal tubular damage [47]. It is contraindicated in patients allergic to penicillin [47]. TETA chelates and promotes urinary Cu excretion and similar to zinc inhibits intestinal Cu absorption through MT induction. Unlike DPA, TETA competes for Cu bound to albumin and does not enter the liver [43]. The most common adverse drug reactions of TETA observed are sideroblastic anemia due to chelation of iron, lupus-like reactions, hemorrhagic gastritis, loss of taste, and skin changes, while no hypersensitivity reactions have been reported [48]. Compared to DPA, it evokes a gentler decoppering, a feature making this drug less prone to triggering deterioration in neurological function [43].

More recently, bc-TTM came into focus as an emerging novel therapeutic appearing to fulfill an unmet need by providing an option for single daily dosing [49]. Compared to other decoppering agents, TTM has an extremely high affinity for Cu, i.e., about 10,000 fold higher than those of DPA or TETA. The K_d_ for binding Cu is K_d_ = 2.32 × 10^−20^ M, while those of DPA and TETA are 2.38 × 10^−16^ and 1.74 × 10^−16^, respectively [46]. It detoxifies Cu by binding a tripartite complex with Cu and protein in both liver and blood and promotes the clearance of that trace element through the biliary route. Past studies have shown that this drug is safe, well-tolerated, and effective in reducing free Cu levels [1]. This drug is notably supposed to improve neurological symptoms and associated disabilities in Wilson’s disease patients because of its extremely high affinity for Cu. Due to this high clinical expectation, it is not surprising that it received Fast Track designation in the U.S. and “orphan status” by the Orphan Drug Act (ODA) in 2011 for the treatment of Wilson’s disease in the U.S. and European Union.

TTM salts, such as the ammonium salt (a-TTM), have been successfully tested clinically since 1991 [50]. The drug has been proposed as an excellent option for the initial treatment of Wilson’s disease patients presenting with neurologic symptoms [51]. However, experimental work in sheep showed that not all TTM is excreted, and further, that Mo is widely distributed and retained in many organs including brain and pituitary gland [52]. Long-term studies further showed that animals became infertile and progressively unthrifty after successful treatment with TTM, most likely because Mo is retained within the brain, pituitary and adrenal glands, provoking toxic endocrinopathy [53].

In an uncontrolled, small longitudinal study that enrolled five Wilson’s disease patients with common neurological symptoms, the administration of a-TTM for 8 or 16 weeks provoked elevation of transaminases in one patient, with complete remission after stopping the therapy [54]. Furthermore, several reports demonstrated that the ammonium formulation is rather unstable, preventing its routine clinical use [18]. As an alternative, bc-TTM was introduced as a second-generation analogue of the ammonium salt. Compared to a-TTM, bc-TTM has superior stability and needs to be given only once or twice daily to be therapeutically effective [17]. It was first tested in a phase I study including 18 patients with advanced solid tumors refractory to conventional therapy [17]. In regard to Wilson’s disease, a first open-label, multicentre, phase 2 study revealed that bc-TTM once-daily given in a dosage of 15 to 60 mg/day over 24 weeks is principal therapeutic promising, but resulted in reversible increased concentrations of asymptomatic alanine or aspartate aminotransferase levels in 39% of all patients that received at least 30 mg/day [1].

As already discussed above, another critical issue is the uptake of Mo. This essential element has a relatively low toxicity in humans and the physiological average dietary intake is about 0.1–0.5 mg Mo. Long-term excessive exposure to this element (i.e., 10–15 mg/d) has been reported to induce adverse side effects including hyperuricemia, arthralgia, erythema, edema and deformation of the knees, hands and feet [55]. Elimination of Mo-containing compounds occurs primarily by excretion through the kidney, while very little of this element is excreted into bile [55]. With regard to bc-TTM, a recent study showed that this formulation is eliminated in a similar fashion to the ammonium salt [19]. Importantly, LEC rats carrying defect *Atp7b* genes and having an abnormal Cu metabolism comparable to human Wilson’s disease showed a lowered excretion of Mo than LEA control rats, most likely because the genetic *Atp7b* defect leads to decreased biliary Mo excretion in addition to the known defect in biliary Cu excretion.

We have previously shown that LA-ICP-MS imaging is well-suited to measure Cu overload in the *Atp7b*^−/−^ mouse model (Appendix A) [21,22,31]. In the present study, we used this methodology to evaluate the therapeutic efficacy of bc-TTM in this Wilson’s disease model. We found that low bc-TTM concentrations (1 mg/kg body weight) provoked increased hepatic Cu content, while the administration of higher concentrations (5 mg or 10 mg/kg body weight) resulted in moderate or 30% removal of hepatic Cu when applying this drug for 4 weeks, (Figure 1). The application of bc-TTM was associated with an increase of lysosomal electro-dense particles that were not seen after the application of DPA or trientine (Figure 5) and were accompanied by a significant serum ALT elevation in the 1 mg bc-TTM group (Appendix A). In line with the proposed biliary excretion route, we found bc-TTM treatment to increase hepatic Mo concentration and serum Mo levels (Figure 4 and Appendix A).

When the treatment regimen was extended to 8 weeks, we observed a significant reduction of hepatic Cu compared to untreated controls (10.91 ± 8.03 µg/g vs. 26.93 ± 30.41 µg/g liver tissue), while TETA or DPA treatment for 8 weeks decreased hepatic Cu concentrations to 7.0 ± 2.88 µg/g or 6.47 ± 5.98 µg/g liver tissue, respectively (Figure 3 and Appendix A). It should be noted that TETA and DPA remove Cu largely through urine, while TTM removes excess Cu via bile, which is the natural route for excess Cu elimination. This physiological route requires that the Cu-TTM complex is first targeted to the liver, explaining why we observed an increase of Cu in the 1 mg/kg body weight group during short term treatment (4 weeks). In line with the proposed excretion route of TTM, we were able to demonstrate the expected increase of hepatic Mo content during bc-TTM treatment, confirming previous reports highlighting extremely important aspects of TTM treatment in experimental models. In this regard, several landmark papers need to be considered to understand the process of TTM-induced Cu elimination. Basically, there are two Cu varieties attracted by TTM, namely, free inorganic Cu and Cu bound to MT. Pioneering work from Jeannin and coworkers showed that inorganic Cu forms heterobimetallic complexes through -Mo-S-Cu clusters [56]. With regard to MT-fixed Cu, TTM has different mechanisms by which to remove Cu bound to MT [11]; for example, it can react selectively and directly with Cu bound to MT, forming MT/TTM complexes, or alternatively, when applied in higher concentrations, it reacts with formed MT/TTM, resulting in the liberation of MT, thereby forming soluble Cu/TTM complexes [11]. Excessive TTM treatment causes a precipitation of an insoluble polymer composed of Cu and Mo [11]. A subsequent study confirming the formation of the insoluble hepatic Cu/TTM polymer showed that TTM is taken up by the liver depending on the amount of Cu bound to MT, while the final efflux of Cu is mediated into the bloodstream together with Mo in the form of Cu/TTM complexes [12]. Based on a comparative analysis of Cu content in LEC and LEA rats after treatment with different TTM concentrations, it was proposed that TTM is incorporated into the liver according to the amount of Cu accumulated in this tissue, while the removal of excess Cu in a form bound to MT is safest when the drug is used as small doses at sufficient intervals [12].

The molecular structure of these particles was further investigated in TTM-treated LEC rats. It was found that the liver lysosomes contained Cu-Mo-S clusters in which Mo was coordinated by four sulfurs with three Cu atoms, while Cu was coordinated to 3–4 sulfurs with approximately one Mo neighbor (Figure 7) [14]. Importantly, these polymetallic clusters were shown to decrease the bioavailability of Cu and, potentially, its redox properties [14].

Most likely, the electron-dense particles we detected in transmission electron microscopy (Figure 5) were deposits of the proposed Cu-Mo-S aggregates, and their occurrence confirmed the proposed process by which TTM evolves in terms of its therapeutic efficacy.

It once again should be emphasized that the formed Cu/TTM complex with a binding constant of K_d_ = 2.32 × 10^−20^ M is extremely stable. The high affinity of TTM to Cu also explains the toxicity of excess dietary molybdate (MoO_4_^2−^) reported in ruminants. Although MoO_4_^2−^ itself has only low affinity for Cu, TTM is formed in the ruminants’ digestive track resulting in severe Cu deficiency [57]. In milking cattle, MoO_4_^2−^ intoxication is linked to a fatal disorder known as “teart” pastures syndrome, which is marked by extreme diarrhea causing marked loss of condition. In sheep, excess MoO_4_^2−^ provokes a degenerative neurological disorder known as “swayback”, resulting from demyelination [57]. Both disorders arise from Cu deficiency, and the symptoms are readily reversed with Cu supplementation [57]. The oral LD_50_ for Mo in rats ranges from 101–330 mg Mo/kg body weight [58], while a lethal repeated oral dose for mouse, guinea pig and rabbit lies between 60–330 mg Mo/kg body weight and levels of 5 mg Mo/kg body weight cause diarrhea, anemia, and skeletal lesions [59].

When comparing Mo-containing compounds, a 10-day dietary exposure to 0–6 mg Mo/kg/day given as a-TTM resulted in a 10% decrease in body weight in rats [60]. Similarly, the administration of 12 mg/kg/day of a-TTM combined with a diet containing 110 mg/kg Cu resulted in lowered erythrocyte counts, hemoglobin, hematocrit, and sperm counts in rats [61].

In our study we applied 1 mg, 5 mg, or 10 mg bc-TTM/kg body weight per oral gavage four times a week for four weeks or 1 mg, or 5 mg bc-TTM for eight consecutive weeks. Previous work in rats using the conversion of animal doses as per US Food and Drug Administration (FDA) guidance, calculated a dose of 1.5 mg/kg to be an equivalent dose of ~0.25 mg/kg corresponding to the typical dose used in man of 30 mg when assuming a body weight of 60 kg and a bioavailability of 50% [19]. 10 mg bc-TTM correspond to 5.18 mg TTM/kg or 2.22 mg Mo/kg body weight, when considering the molecular weights of bc-choline TTM (C_10_H_30_MoN_2_O_2_S_4_; 434.54 g/mol), TTM (H_2_MoS_4_^2−^, 224.21 g/mol), and Mo (95.95 g/mol), respectively. So the highest content one mouse received during our treatment regimen (32 applications, 10 mg bc-TTM/dose) about 166 mg TTM or 71 mg Mo. In comparison, in the phase II study of Weiss et al. [1], bc-TTM given once daily in a dosage of 15 to 60 mg/day would correspond to 0.19 to 0.75 mg/kg body weight (assuming a body weight of 80 kg, i.e., it would be comparable to a dose of between 1 mg and 5 mg in our mouse experiment. Similar to DPA and TETA, this treatment resulted in a significant decrease in hepatic Cu without the occurrence of any drug adverse effects, while untreated animals had significantly higher hepatic Cu content and frequently developed tumors. The tumors observed in one animal in the 5 mg treatment group might have resulted from the fact that the therapeutic treatment was initiated late, i.e., at age 36 weeks. *Atp7b*^−/−^ mice older than 28 weeks develop gross anatomic liver abnormalities that progress into tumors [20]. Possibly, earlier treatment would have been even more beneficial in our experiments.

It should be noted that the dosage of 10 mg/kg body weight was more than double that used in the phase II study in Wilson’s disease patients [1]. Such a concentration of bc-TTM triggers ultrastructural mitochondrial lesions, which are typically characterized by the formation of holes of variable size (Figure 6). It is well known that TTM is an effective inorganic hydrogen sulfide (H_2_S) releasing agent [62]. H_2_S is readily water soluble, forming high quantities of hydrogen sulfide ions (HS^−^) which bind and inhibit cytochrome C oxidase in complex IV of the electron transport chain [63]. As such, it provokes considerable oxidative stress that might be reflected in mitochondrial damage. Although these findings may not be applicable to patients with Wilson’s disease, the data suggest that the drug should be administered at low concentrations to prevent mitochondrial dysfunction and overload of hepatic excretory pathways.

In humans, a phase 2 study showed that bis-TTM is effective in improving neurologic manifestations of Wilson’s disease, suggesting this drug to be particularly suitable for neurologic-predominant Wilson’s disease [1,64]. Unfortunately, we have not yet analyzed the impact of bis-TTM on cerebral Cu content and distribution. Such measurements will be of fundamental importance to finally evaluate the therapeutic efficacy in our experimental model. In the past, we used LA-ICP-MS technology to analyze the regional distribution of cerebral Cu overload in *Atp7b*^−/−^ mice, and demonstrated that genetic long-term correction by AAV8-based therapy is beneficial in reducing the overall concentration of total cerebral Cu in this model [21,33].

TTM and Cu form an extremely stable complex (K_d_ = 2.32 × 10^−20^ M), in which Mo is integral part of TTM [46]. Therefore, comparative imaging of Mo in samples taken from untreated and TTM-treated mice makes it possible to indirectly imaging the Cu/TTM complex. In addition, the stability of the complex prevents the release of free Mo that, in free form, leads to anemia, anorexia, profound diarrhea, joint abnormalities, osteoporosis, hair dislocation, reduced sexual activity and death [65]. In line with these facts, previous results from a phase II trial showed that bc-TTM has an overall favorable safety profile [1,66].

One limitation of our study was the usage of mice at different ages during our dose determination and therapeutic treatment studies. It is well-known that Cu accumulation in younger animals, in particular in the chosen animal model, is different than that in older animals [20]. However, general therapeutic aspects of decoppering drugs can be tested throughout the life span in these and other Wilson’s disease models.

In future work, we will use the *Atp7b*^−/−^ mouse model and preserved materials of the presented study to delineate the impact of bc-TTM on cerebral Cu content, and to determine if this Cu-chelating therapy interferes with homeostasis of other elements. We have previously shown that the concentrations of other metals, including iron, zinc, manganese, sodium, magnesium, potassium, calcium, phosphorus, chromium, nickel and lead, were unaffected after gene restoration in *Atp7b*^−/−^ mouse [21]. However, we have also shown that Cu overload in cortex (grey matter), corpus callosum (white matter) and cerebellum is associated with significantly zinc overload, while other elements such as iron and manganese were not altered in the respective brain areas [33]. However, element imaging in brain is more complex than in liver, in which elements are more homogenously distributed. Data from a previous study of our laboratory, in which we imaged Cu in murine brains by LA-ICP-MS, revealed that cerebral Cu concentrations are highly variable, showing a distinctive spatial distribution [33]. Although the Cu content in the entire section of a murine brain is about 4.2 ± 1.4 µg/g, the content ranges from 1.5 ± 0.5 µg/g in the corpus callosum to 6.6 ± 1.6 µg/g in the cerebellum. In addition, we have already noted that the concentration of a particular element in brain sections strongly depends on the sectional plane analyzed, making the preparation of comparable specimens reflecting the same regions and planes more complicated [33].

It should be noted that Cu overload does not only play an important role in the pathogenesis of Wilson’s disease. Copper chelating therapy is also of potential use in terms of interfering with the pathogenesis of Alzheimer’s and Parkinson’s disease, in idiopathic pulmonary fibrosis, diabetes and in many form of cancers [67,68,69]. Therefore, it will be interesting to test the therapeutic effect of bc-TTM in these disease models.

## 5. Conclusions

In summary, the oral administration of TTM in *Atp7b*^−/−^ mice led to marked reduced hepatic Cu. However, the mechanism of this decoppering drug seems to be more complex than those of DPA or TETA. Based on the route of Cu removal, the application of small doses for a short time initially provoked increased hepatic Cu that was most likely fixed to Mo-containing complexes. Higher doses of TTM or prolonged treatment intervals resulted in a marked reduction of hepatic Cu. However, based on the reported toxicity of Mo when applied in excess and the discussed removal mechanism that is different from those of DPA and TETA, we believe that TTM as a decoppering agent is safest when used at small doses and applied as early as possible.

## Figures and Tables

**Figure 1 biomedicines-09-01861-f001:**
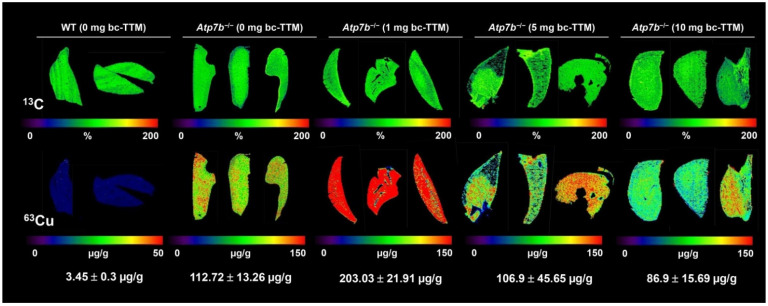
Dose finding study for tetramolybdate (bc-TTM) in lowering hepatic copper content as assessed by LA-ICP-MS. Liver sections were prepared from *Atp7b*^−/−^ mice that received different doses of bc-TTM. The specimens were imaged for carbon (^13^C) as a reference and copper (^63^Cu). Representative images of three animals per group and calculated mean concentrations ± SD of Cu are depicted. Copper concentrations are depicted in a scale from 0 to 150 µg/g liver tissue. Liver section from wild type controls depicted in a scale from 0 to 50 µg/g liver tissue had a Cu concentration of 3.45 ± 0.3 µg/g liver tissue.

**Figure 2 biomedicines-09-01861-f002:**
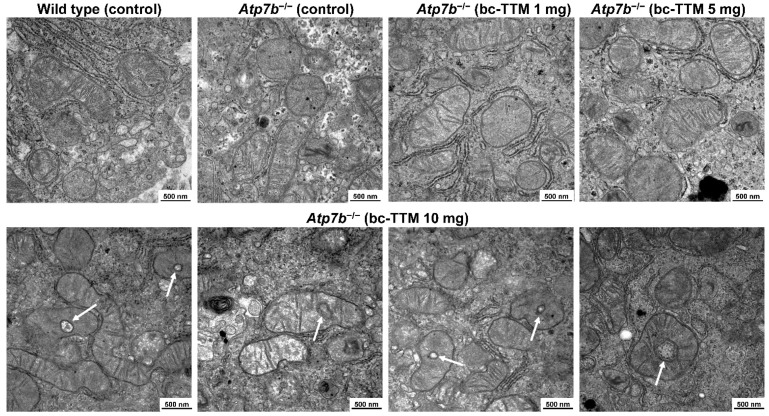
Transmission electron microscopic images of mitochondria in liver sections after therapeutic treatment with bis-choline-tetrathiomolybdate. Representative images of hepatic mitochondria after bc-TTM treatment are depicted. As controls, mitochondria from untreated wild type and *Atp7b*^−/−^ mice are depicted. Lumina in mitochondria in the 10 mg bc-TTM group are marked by white arrows. Space bars represent 500 nm.

**Figure 3 biomedicines-09-01861-f003:**
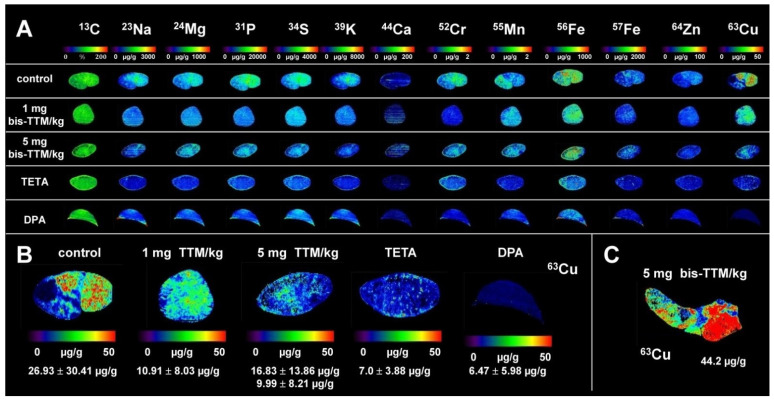
LA-ICP-MS imaging in liver sections of *Atp7b*^−/−^ mice treated with decoppering agents. (**A**) Simultaneous LA-ICP-MS imaging of carbon (^13^C), sodium (^23^Na), magnesium (^24^Mg), phosphorus (^31^P), sulfur (^34^S), potassium (^39^K), calcium (^44^Ca), chrome (^52^Cr), manganese (^55^Mn), iron (^56^Fe, ^57^Fe), copper (^63^Cu), and zinc (^64^Zn) was done in cryo-cuts from *Atp7b^−/−^* mice treated with indicated decoppering agents. *Atp7b*^−/−^ mice that received water served as controls. Element concentrations are given in µg/g liver tissue, except C used for standardization given in %. (**B**) Comparison of Cu measurements depicted in (**A**). The determined hepatic Cu concentrations (Mean ± SD) of the different treatment groups (*n* = 5 animals per group) are shown. In the 5 mg TTM/kg group, the lower values for mean ± SD refer to the calculation when only four animals were considered. (**C**) In the 5 mg bc-TTM/kg body weight, one animal developed tumors highly enriched in Cu. Abbreviations used are: bc-TTM, bis-choline-tetrathiomolybdate; DPA, D-penicillamine; TETA, trientine.

**Figure 4 biomedicines-09-01861-f004:**
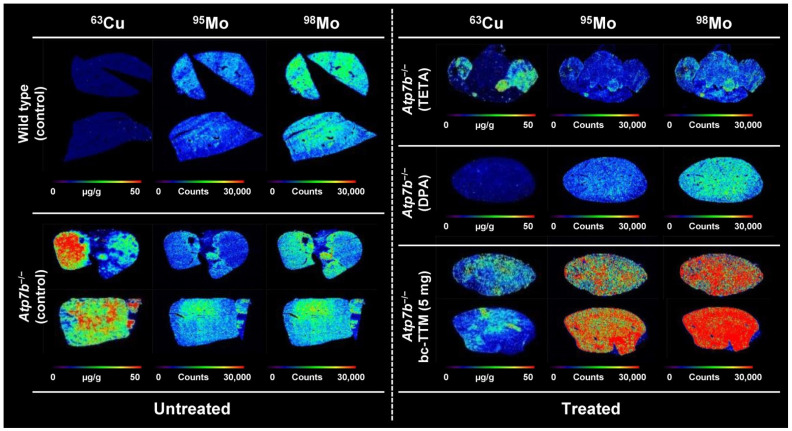
Tissue distribution of copper and molybdenum in liver after therapy with bis-choline-tetrathiomolybdate. Liver sections were prepared from *Atp7b*^−/−^ mice that received indicated drug regimens. Untreated wild type or *Atp7b*^−/−^ served as controls to estimate the therapeutic efficacy of individual drugs. The specimens were imaged for copper (^63^Cu) and molybdenum (^95^Mo and ^98^Mo). Copper concentrations are depicted in a scale from 0 to 50 µg/g liver tissue, while Mo is given in counts. Abbreviations used are: *Atp7b*^−/−^, mouse deficient for the *Atp7b* gene; bc-TTM, bis-choline-tetrathiomolybdate; DPA, D-Penicillamine; TETA, trientine.

**Figure 5 biomedicines-09-01861-f005:**
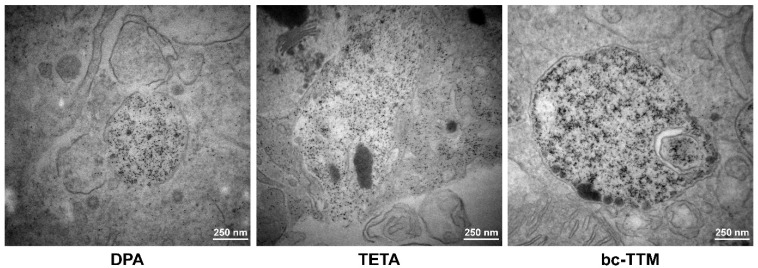
Comparative analysis of liver pieces from *Atp7b*^−/−^ mice treated with different decoppering agents. Representative liver sections from *Atp7b*^−/−^ mice treated with D-penicillamine (DPA), trientine (TETA) or bc-tetrathiomolybdate (bc-TTM) were fixed and examined by electron microscopy. The bc-TTM-treated mice showed more electron-dense lysosomal deposits than those treated with DPA or trientine. Space bars represent 250 nm.

**Figure 6 biomedicines-09-01861-f006:**
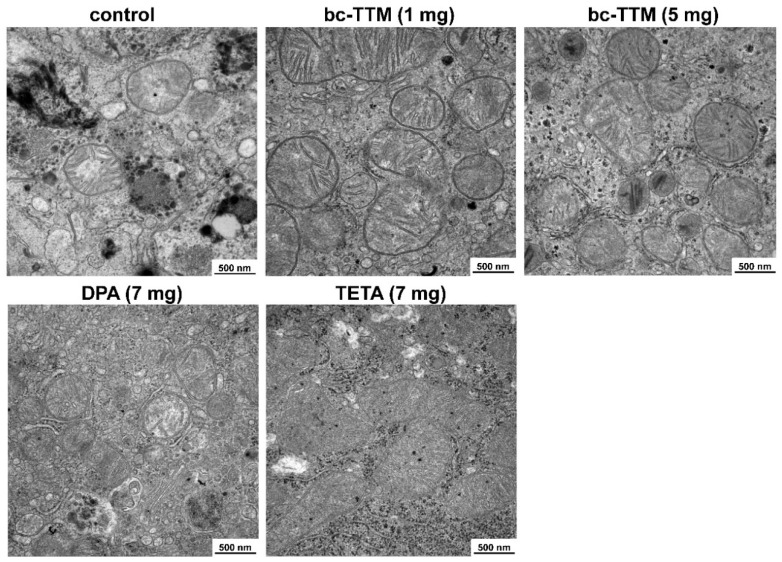
Transmission electron microscopic images of mitochondria in liver sections after therapeutic treatment with bis-choline-tetrathiomolybdate (bc-TTM). Representative images of hepatic mitochondria after bc-TTM treatment are depicted. As controls, normal mitochondria from untreated *Atp7b*^−/−^ mice, as well mitochondria from *Atp7b*^−/−^ mice that received D-Penicillamine (DPA) or trientine (TETA) are depicted. Space bars represent 500 nm.

**Figure 7 biomedicines-09-01861-f007:**
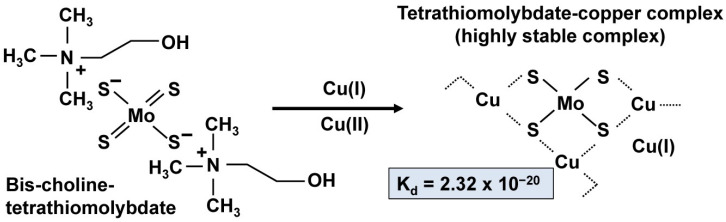
Proposed structure of the bis-choline tetrathiomolybdate-copper complex. Bis-choline tetrathiomolybdate and Cu form a highly stable complex with a K_d_ = 2.32 × 10^−20^ M). In the proposed model, each molybdenum is fixed by four sulfurs that also have affinity for Cu(I) ions.

## Data Availability

The data presented in this study are available on request from the corresponding authors.

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
