# Peer review of "Analyzing the Therapeutic Efficacy of Bis-Choline-Tetrathiomolybdate in the Atp7b−/− Copper Overload Mouse Model"

_biomedicines, 2021, doi:10.3390/biomedicines9121861_

Round 1
Reviewer 1 Report
In this manuscript the authors studied the effect of bc-TTM treatment on hepatic Cu content in Atp7b-/- mice (an experimental model of Wilson disease). The authors found that treatment of Atp7b-/- mice with bc-TTM (1 mg/kg b.w.) four times a week for four consecutive weeks induced elevation of Cu content in liver compared to the corresponding controls. This effect was not observed for higher doses of the chelating agent (5 and 10 mg/kg b.w.). The authors established that elevation of copper content was associated with a significant increase of zinc and iron hepatic contents. A significant elevation of serum ALT activity was also observed for the dosage 1 mg/kg b.w. of bc-TTM. The authors explained the effect of 1 mg/kg b.w. bc-TTM on hepatic copper with the excretion route of Cu-TTM complex. Prolonged treatment regimen (8 weeks) with the chelating agent resulted in a significant reduction of hepatic copper content. The chelating agent bc-TTM was even more effective compared to TETA and DPA in mobilizing hepatic copper concentration.
To my opinion this study was well conducted. The conclusion corresponds to the experimental results. The authors discussed the available relevant literature data in details. Therefore, I recommend the publication in the manuscript in Biomedicines.
Author Response
Dear reviewer,
please find our comments to your report attached.

Reviewer 2 Report
The paper entitled “Analyzing the therapeutic efficacy of bis-choline- tetrathiomolybdate in the Atp7b-/- copper overload mouse model” provides new knowledge concerning the effects of bis-choline-tetrathiomolybdate (bc-TTM) on physiological characteristics in the Atp7b-/- copper overload mouse model, and may contribute to improve Wilson disease treatments in the future. I read the manuscript with interest, and I think that this paper may be had interest by researchers who study Wilson disease treatments. However, there are some significant concerns in this paper. Concerns raised are shown as below.
Major concerns:
1) Figure 1
The authors described "Unexpectedly, the treatment with bc-TTM in the 1 mg group caused doubling of hepatic Cu concentration from 112.72 ± 13.26 μg/g liver tissue to 203.03 ± 21.91 μg/g liver tissue, while a moderate therapeutic effect in the 5 mg group (106.9 ± 45.65 μg/g) and an 30% reduction of this trace element in the 10 mg TTM group (86.9 ± 15.69 μg/g) was observed." Are these data statistically significant? How did the authors calculate "30% reduction"?
2) Figure 3
The authors described "During therapy with 1 mg bc-TTM, the Cu concentration decreased from 26.93 ± 30.41 μg/g to 10.91 ± 8.03 μg/g liver tissue, while TETA or DPA lowered hepatic Cu concentration of 7.0 ± 2.88 μg/g or 6.47 ± 5.98 μg/g liver tissue (Table S3)". Are these data statistically significant? I think these data are not suitable for evaluating the effects of bc-TTM on Cu accumulation in liver because there is great variability between them. Therefore, the authors should obtain reliable data using enough the number of mice in this experiment.
3) Figure 4
I think that the authors should explain the physiological or pathological significances of elevated hepatic Mo concentrations for the 5 mg bc-TTM group in more detail.
Minor concerns:
1) I think that the authors should explain the reason why the authors also mentioned to 65Cu in addition to 63Cu (p.8, line 15).
2) I think the "Discussion section" is too long.
3) I cannot see the part of supplemental tables.

Author Response

(The authors gave the same response as above.)

Reviewer 3 Report
The manuscript “Analyzing the therapeutic efficacy of bis-choline-tetrathio-molybdate in the Atp7b -/-
copper overload mouse model” by Kim et al. is a research article in which the authors describe the
comparison of copper chelation therapy using three different agents (namely ALXN1840, D-penicillamine
and trientine) in lowering hepatic copper content in Atp7b -/- mouse.
The study, supported by Alexion Pharmaceuticals, is aimed to test the effect of bc-TTM, a more stable and
amenable for clinical use version of TTM, in the murine model of Wilson disease.
General comment:
Authors build on the extensive experience of the Dr. Ralf Weiskirchen’s group on liver diseases including
Wilson disease. In particular, authors have previously described the technique of LA-ICP-MS for
simultaneous imaging and quantification of a variety of trace metals.
High standard deviations in Cu measurements upon different treatments shown in Fig 3 denote high inter
group variability, possibly due to technical or biological reasons. Authors should provide a scatterplot
analysis of all copper values and formal statistical analysis between groups.
Limitation
As authors clearly stated: “these experimental results may not be applicable to patients with Wilson
disease”.
Minor issues:
1. Abstract: even if the abbreviation is quite obvious, it is advisable to provide the “bc-TTM”
abbreviation near the full form before using it.
2. Section 2.1: “Bosten”, revise.
3. Section 2.3 title: “Ttreatment”, revise.
4. Section 2.3. Briefly motivate the rationale to use mice of different ages for the dose finding and
the therapeutic efficiency experimental arms. Copper accumulation in younger animals is
higher. Consequently also the extent of treatment-related removal of copper accumulated in
tissues in young animals might be different to the one observed in older mice.
5. Section 2.6 title: “ICP-MSMmeasurements”, revise
6. Section 2: Materials and Methods. How statistical analysis was performed? Describe in
methods and indicate in figure 1 and 3.
7. Section 3.1 title: “Mmice”, revise
8. Section 3.1: “Most strikingly, Cu content was dramatically lowered in the regenerative
nodules”. Please highlight also in Figure S1 panel A the nodules you are referring to. Add scale
bars in Figure S1
9. Fig S1: “Light microscopy of representative unstained liver sections from 10-month-old wild
type and Atp7b-/- mice”. Check figure order on panel A.
10. Check supplementary tables size to make them fully readable. As presented they cannot be
used for revision. The same for fig S3.
11. Section 3.3.: “10 mg TTM group” “1 mg TTM group”. To avoid confusion consistently use the
notation bc-TTM
12. Figure 3: “In the 5 mg TTM/kg group, the lower values for mean +/- SD refer to the calculation
when only four animals were considered”. Please, provide as supplementary figure, a
scatterplot analysis of all copper values for the 5 animals for each group.
13. Figure 5: “The bc-TTM-treated mice showed more electron-dense lysosomal deposits than
those treated with DPA or trientine”. Please, in addition to qualitative analysis on one single
specimen per group presented in the figure, provide additional quantitative data to support
this statement.
14. Page 14: “the data suggests that the drug should be when possible administered”. Revise.
15. Page 14: WD. Use Wilson disease as in all manuscript.
16. Page 14: “Unfortunately, we have not analyzed the impact of bis-TTM on cerebral Cu content
and distribution yet”. Any plan to assed also brain Mo accumulation?
17. Copper chelation therapy has been proposed to counteract a variety of disorders, including
BRAF V600E cancers (Baldari et al., 2019; Brady et al., 2017). Do you envision a possible use of
bc-TTM other than for Wilson disease?
18. “All authors have read and agreed to the published version of the manuscript”. Nonetheless,
there are 3 typos in 3 different sub-heading titles. To avoid this kind of nuisance I strongly
encourage the authors to actually (re)-read their manuscript before submission. Another
suggestion to make reviewer’s job easier is to use the journal template with lines numbers as
requested.
References:
Baldari, S, Di Rocco, G, Heffern, MC, Su, TA, Chang, CJ, Toietta, G (2019) Effects of copper chelation on
BRAF(V600E) positive colon carcinoma cells. Cancers 11: 17.
Brady, DC, Crowe, MS, Greenberg, DN, Counter, CM (2017) Copper chelation inhibits BRAF V600E-driven
melanomagenesis and counters resistance to BRAF V600E and MEK1/2 inhibitors. Cancer Res 77:
6240-52.

Author Response

(The authors gave the same response as above.)

Round 2
Reviewer 2 Report
My questions and comments on the manuscript are now appropriately responded.